# Recent Advances in Molecular and Cellular Functions of S100A10

**DOI:** 10.3390/biom13101450

**Published:** 2023-09-26

**Authors:** Gillian C. Okura, Alamelu G. Bharadwaj, David M. Waisman

**Affiliations:** 1Department of Pathology, Dalhousie University, Halifax, NS B3H 1X5, Canada; gillian.okura@dal.ca (G.C.O.); alamelu.bharadwaj@dal.ca (A.G.B.); 2Departments of Biochemistry and Molecular Biology, Dalhousie University, Halifax, NS B3H 1X5, Canada

**Keywords:** S100A10, Ca^2+^-binding, fibrinolysis, cancer, depression

## Abstract

S100A10 (p11, annexin II light chain, calpactin light chain) is a multifunctional protein with a wide range of physiological activity. S100A10 is unique among the S100 family members of proteins since it does not bind to Ca^2+^, despite its sequence and structural similarity. This review focuses on studies highlighting the structure, regulation, and binding partners of S100A10. The binding partners of S100A10 were collated and summarized.

## 1. Introduction

S100A10 (p11) is one of twenty-five different S100 family members of proteins, of which most are genes located on chromosome 1q21, which is amplified in several neoplastic diseases. Although the S100 proteins demonstrate high-level sequence and structural similarity, they are markedly different in their functions and the proteins they interact with. S100A10 is unique among the twenty-five S100 proteins as it does not bind to Ca^2+^ and is present in a conformation that resembles the Ca^2+^-bound form of other S100 proteins, suggesting it perpetually exists in the active state. S100A10 interacts with several proteins but predominantly exists in a heterotetrameric complex with Annexin A2 (p36), which plays an important role in the cellular localization, stability, and functionality of S100A10. Over the past decade, there have been several excellent reviews that have been published on S100A10. Our focus in this review is to highlight the recent advances in the structure, regulation, binding partners, interactions, and functions of S100A10.

## 2. Structure

The first demonstration of the importance of calcium (Ca^2+^) as a signaling molecule was the evidence that Ca^2+^ played a key role in muscle contraction. This discovery was made by Sidney Ringer, who reported that “phosphate of calcium added to saturation to saline can sustain contractility [1]. Subsequently, Heildebrun reported that “calcium, even in rather high dilution, causes immediate and pronounced shortening of frog muscle”, and that “this effect is not shared by any one of the other cations normally present in any quantity in muscle, i.e., K^+^, Na^+^, and Mg^2+^” [2]. Bailey also envisioned the importance of Ca^2+^ ions in muscle contraction and reported that actomyosin ATPase was stimulated by Ca^2+^ and, to a much lesser extent, Mn^2+^ [3]. This work was corroborated by Bozler, who demonstrated that the removal of Ca^2+^ by the chelator, EDTA, relaxed muscle fibers [4]. Subsequently, Ebashi reported that Ca^2+^, at a micromolar concentration, caused the contraction of actomyosin preparations [5]. Ebashi later revealed that the receptor that mediated Ca^2+^-dependent regulation of muscle contraction was a protein he named troponin. Therefore, troponin was the first intracellular Ca^2+^-binding protein identified [6]. Later, it was shown that a subunit of troponin, troponin-C, was the Ca^2+^-binding protein and that the binding of Ca^2+^ to troponin-C induced a conformational change in troponin that resulted in the activation of actomyosin ATPase. This established that Ca^2+^ and a receptor of Ca^2+^, namely a Ca^2+^-binding protein, regulated muscle contraction.

An understanding of how proteins bound Ca^2+^ was provided by Kretsinger, who first reported the X-ray crystallographic structure of the carp muscle Ca^2+^-binding protein, parvalbumin [7]. He demonstrated that parvalbumin possessed a Ca^2+^-binding domain that he referred to as an EF-hand. The designation “EF-hand” was derived from the structural orientation of the two α-helices (E and F) that form together with the Ca^2+^-binding loop of 12 amino acids. The EF-hand motif is the most frequently occurring Ca^2+^-binding motif in eukaryotic systems [8].

There are more than 66 subfamilies of EF–hand proteins [9], which include well-known families such as the calmodulin (CaM) and S100 families. The conformational change induced by the binding of Ca^2+^ to each of these families is fundamentally different. The Ca^2+^-induced conformational change in all S100 proteins results from a shift in the orientation of Helix III [10], whereas the conformational change in calmodulin involves a dramatic opening of both EF-hands, which is not observed in the S100 proteins. Furthermore, calmodulin utilizes both EF-hand domains to bind to a target molecule, whereas an S100 dimer binds two target molecules on opposite faces of the structure. CaM has two independent globular domains tethered by a flexible linker, which enables the two binding sites to bind to the same target. In contrast, the S100 dimer forms a single, highly interdigitated globular domain, with the two symmetrically disposed binding sites on opposite faces of the structure. Like other EF-hand proteins, the conformational change results in the exposure of a hydrophobic patch that serves as the key factor driving the binding of targets [9,10,11]. In contrast, a unique dimeric structural organization has been observed for the S100A6/RAGE receptor complex, where the two symmetrical binding sites for the RAGE receptor are localized to the same face of the S100A6 dimer [12].

The S100 protein family is the largest subfamily of EF-hand proteins. They were initially identified for their solubility in 100% ammonium sulfate [13]. This was true, however, for the first two proteins discovered, namely S100B and S100A1, but it is no longer a shared feature among the twenty-three S100 members identified so far in humans, which include S100A1, S100A2, S100A3, S100A4, S100A5, S100A6, S100A7, S100A7A, S100A7B, S100A8, S100A9, S100A10, S100A11, S100A11P, S100A12, S100A13, S100A14, S100A15, S100A16, S100A17, S100A18, S100B, S100G, S100P, and S100Z. The S100 proteins are abundant, low molecular weight (10–12 kDa), acidic proteins exhibiting distinct tissue-specific expression. All, except S100G, exist as dimers, particularly but not exclusively homodimers, under physiological conditions. The dimer is formed by noncovalent interactions between helices I and IV. NMR, mass spectrometry, and Green Fluorescent Protein (GFP) trap experiments consistently show that S100A1:S100B, S100A1:S100P, and S100A11:S100B heterodimers are the predominant species formed compared to their corresponding homodimers [14,15,16].

The S100 monomers are typically composed of an EF-hand motif at the amino-terminal (EF1) and one at the carboxyl-terminal (EF2) connected by a region referred to as a “hinge”. The EF1 motif contains a long helix (HI), a loop (L), and a short helix (HII). The EF1 Ca^2+^-binding site is called a ‘pseudo’ EF-hand because it possesses two extra residues in the loop, and the coordination of Ca^2+^ is accomplished mainly by backbone carbonyls. In contrast, the canonical EF2 Ca^2+^-binding site contains a short helix (HIII), a loop (L), and a long helix (HIV) and binds Ca^2+^ through acidic side chains (Figure 1).

As discussed, Ca^2+^-binding to the S100 proteins exposes a hydrophobic region comprising residues from both helix III and IV, which serves to bind target proteins. Structural studies have established that Ca^2+^-binding to the S100 protein, except S100A10, is required to facilitate most S100-annexin interactions [17]. Of the twenty-five members of the S100 protein family, seven (S100A1, S100A4, S100A6, S100A10, S100A11, S100A12, and S100B) interact with at least one of the 12 human annexin proteins. S100 proteins, such as S100A6, for example, appear to form complexes with several annexin proteins (A2, A5, A6, and A11) [18,19] (Figure 2).

Annexins are a family of Ca^2+^-binding proteins that interact with biological membranes. There are twelve human annexins. All annexins have four repeating, highly homogenous amino acid sequences (eight repeats in the case of annexin VI), known as the annexin repeat [20] (Figure 3). The annexins typically have three different Ca^2+^-binding sites distinct from the EF-hand sites. The type II Ca^2+^-binding sites have the canonical sequence of the endonexin fold, KGXGT-(38X)-D/E, a Ca^2+^-binding site characteristic of the annexin protein family. The annexins are noted for a single type II site in the second, third, and fourth domains, two type III Ca^2+^-binding sites in the first domain of the protein, and a type AB’ site [21]. Annexin A2 has been extensively studied and shown to bind to multiple proteins, as well as phospholipids [22], polysaccharides such as heparan [23], fucoidan [24], and RNA [25] (reviewed in [26,27]). Specific and well-established binding sites on annexin A2 include residues K279 and K281, which form a phosphoinositide-specific binding site that contributes significantly to the specificity of annexin A2 binding to phosphoinositide-containing membranes [28], and the last nine residues of the carboxyl-terminal, LLYLCGGDD, that form the F-actin binding site [29].

S100A10 differs from the others as it cannot bind Ca^2+^ ions because of a three-residue deletion in EF-1 and the mutation of acidic Ca^2+^-coordinating residues in EF-2. However, S100A10 is virtually identical to Ca^2+^-loaded S100 proteins, suggesting S100A10 has adapted an active conformation [17]. Multiple laboratories have studied the AIIt complex in great detail (reviewed in [26]). AIIt is a highly symmetric complex with the S100A10 antiparallel dimer in the center and an annexin A2 monomer on either side of S100A10. The N-terminus of each annexin A2 monomer is associated with a region formed by EF1-Helix I from one S100A10 monomer and the EF2-Helix IV and hinge from the other S100A10 monomer. Recently, the Ca^2+^-free and seven Ca^2+^-loaded AIIt complexes have been analyzed by molecular dynamics [30]. The advantage of this approach was that the modeler added the last five residues of S100A10, Q93, K94, G95, K96, and K97, which were absent from the crystal structure. These authors reported that, consistent with previously published data, the N-terminal annexin A2 hydrophobic residues (T3, V4, I7, L8, and L11) form contacts with multiple residues of S100A10 and are the top ten strongest interactions for both annexin A2 in the Ca^2+^-free and Ca^2+^-loaded models. These authors found that while the S100A10 residues (E6, M9, E10, M13, F14, F42, A82, and Y86) formed strong contacts with the N-terminal hydrophobic residues of annexin A2, five residues (F39, L79, C83, F87, and M91), thought to be important complex-forming residues, provided only weak interactions, suggesting they have a less significant contribution toward the total interaction between S100A10 and annexin A2.

**Figure 3 biomolecules-13-01450-f003:**
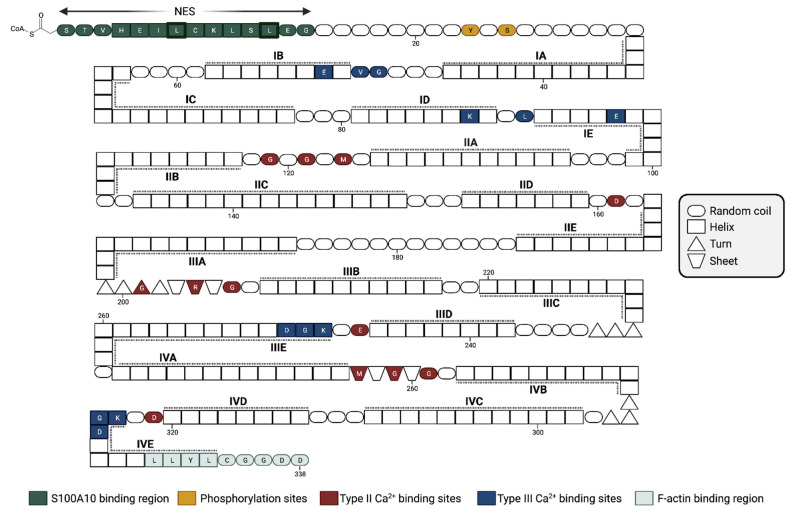
Amino acid structure of annexin A2. Due to high sequence conservation, human annexins are structurally similar, with four annexin repeats (eight for annexin A6). The ~70-residue-long structural element (labeled domains I–IV) consists of five α-helices (labeled A–E) and is important for Ca^2+^-binding. Additionally, other important structural sites include the nuclear export sequence (NES) and multiple phosphorylation sites. Figure updated from [31]. Created with “Biorender.com (accessed on 8 September 2023)”.

It was particularly interesting to note that in the Ca^2+^-loaded model, the annexin A2 residue, R145, interacted with the S100A10 residue, D48, but that in the Ca^2+^-free model, the A2 residue, R145, interacted with the carboxyl-terminal S100A10 residue, K97 [30]. We had initially proposed that K97 was the residue that interacted with plasminogen [32]. This was substantiated by our report that several carboxypeptidases removed this residue, which resulted in the loss of the ability of S100A10 to bind and activate plasminogen [33,34]. However, more recently, we observed that the substitution of the carboxyl-terminal S100A10 residue with isoleucine did not affect plasminogen activation, and we subsequently proposed that the carboxyl-terminal S100A10 residue might actually regulate the exposure of an internal lysine residue [35]. We have also reported that Ca^2+^ did not stimulate S100A10-dependent plasminogen activation [36]. The report that the accessibility of carboxyl-terminal lysine of the S100A10 (K97) was affected by Ca^2+^ further suggested that the carboxyl-terminal lysine of S100A10 might not be critical for plasminogen activation.

Recently, we proposed that the formation of the AIIt complex was an example of a mutualistic symbiotic relationship [27]. In this regard, we suggested that the formation of the AIIt complex not only regulates the biological activity of each subunit but also confers new biological functions that were not possessed by either subunit. Furthermore, complex formation is also critical for the stability of S100A10 because, in the absence of annexin A2, S100A10 is rapidly degraded [37]. Similarly, annexin A2 requires S100A10 to increase its affinity for Ca^2+^, facilitating its participation in Ca^2+^-dependent processes such as membrane binding. These remarkable properties of the annexin A2/S100A10 complex are discussed in detail in our review [27].

The fundamental property that separates S100A10 from the other S100 proteins is that S100A10 cannot bind Ca^2+^ but is locked in the Ca^2+^-loaded conformation. The key question is what advantage the inability to respond to fluctuations in Ca^2+^ confers on the protein. The simplest explanation is that S100A10 will form a complex with annexin A2 in the presence or absence of Ca^2+^, whereas the formation of annexin complexes by other S100 proteins will require Ca^2+^. The ubiquitin-mediated degradation of newly formed S100A10 is Ca^2+^-independent, and therefore the complex formation will be finely tuned to the availability of annexin A2 and not to intracellular Ca^2+^ concentrations. However, any Ca^2+^-dependent interactions of the annexin A2/S100A10 complex with binding partners will be regulated by Ca^2+^-binding by the annexin A2 component of the complex. Since the binding of S100A10 to annexin A2 increases the affinity of annexin A2 for Ca^2+^, it could be argued that free annexin A2 and annexin A2 bound to S100A10 are conformationally distinct. In contrast, considering the micromolar Ca^2+^ binding affinity of most of the S100 proteins, their ability to form complexes with annexins would appear to be unlikely under resting submicromolar intracellular Ca^2+^ concentrations. The exception is S100A1, which, upon covalent modification, dramatically increases its Ca^2+^-binding affinity one-hundred-fold [38].

## 3. Binding Partners of S100A10

In addition to annexin A2, a number of additional proteins have been shown to interact with S100A10. For most of these protein–protein interactions, the physiological significance remains unknown. Additionally, it is unclear if S100A10 is the only protein in the S100 family that will interact with these proteins. A list of published S100A10 protein interactors was generated by summarizing public databases: NCBI (https://www.ncbi.nlm.nih.gov/), BioGRID (https://thebiogrid.org/) and ELIXIR (https://www.ebi.ac.uk/intact/home). Ontology analysis of the protein interactors was performed using DAVID 7.0 [39] (http://david.ncifcrf.gov/). Significant ontology findings with a *p* value < 0.05 and FDR < 0.1 are summarized in Table 1 and illustrated in Figure 4. Interactions that have been demonstrated to have physiological relevance are presented in Table 2. A full list of reported interacting proteins is summarized in Table 3.

To date, over 170 proteins have been proposed to interact with S100A10 and observed through various biochemical approaches. Mass spectrometry was the most commonly used technique to identify S100A10 interactors. This approach involved several techniques, including affinity capture (34% of identified interactors), proximity labeling (13%), and co-fractionation (13%). Two-hybrid yeast screening was the second most popular biochemical approach to identify interactors (13%). Other methods used to identify protein interactors included, but were not limited to, co-crystal structure, affinity capture-western blot, reconstituted complex, co-immunoprecipitation, and fluorescent imaging. It will be interesting to determine the physiological relevance of the predicted binding partners of S100A10 and, more specifically, to identify the binding regions or structural determinants in S100A10 that predict and demonstrate the binding affinity to the various proteins.

## 4. Regulation of S100A10

The expression and function of S100A10 are tightly regulated by various cytokines, growth factors, nitric oxide (reviewed in [69]), RAS oncogene [70], PML-RAR oncogene [71], and hypoxia-inducible factor (HIF) [72]. MicroRNA (miRNA) and long-non-coding RNA (LncRNA) are important non-protein-coding RNA machinery that make up a large percentage of the genome. They have been demonstrated to function in post-transcriptional and epigenetic regulation. LncRNA and miRNA play a crucial role in regulating development, stress-mediated processes, immune response and tumor progression, metastasis, and chemoresistance [73,74,75,76]. Most evidence of the regulation of S100A10 by miRNA and LncRNA has been obtained from studies in hepatocellular carcinoma (HCC) models. Shan et al. identified that mir-590-5p is decreased in some hepatocellular carcinoma cell lines with a concomitant upregulation of S100A10. They further showed that mir-590-5p regulated S100A10 expression by interacting with the 3′UTR of S100A10 mRNA [77]. Similarly, LINC00174 has been shown to be an oncogenic LncRNA in HCC, and it functions by sponging mir-320, which further upregulates the expression of S100A10, accelerating the proliferation and metastasis of HCC cells [78]. Another lncRNA, KCNMB2-AS1, functions by regulating S100A10 in bladder cancer through direct sponging of miR-374a-3p, which negatively regulates the expression of S100A10 [79].

## 5. Role of p11 in Fibrinolysis

S100A10 is present at the endothelial cell surface and bound to annexin A2, forming the AIIt complex. This complex has been shown to be one of the key receptor-binding plasminogens in endothelial cells. Plasminogen (Pg) is an inactive zymogen that is converted to the active serine protease plasmin (Pm) by plasminogen activators, tissue-plasminogen (tPA) activators, and urokinase plasminogen activator (uPA). This process is highly regulated; the principal inhibitor of Pm is α2-antiplasmin (AP), and the principal inhibitor of tPA and uPA is plasminogen activator inhibitor type 1 (PAI-1) [80,81,82,83]. Although tPA and uPA possess low activity, their activity dramatically increases when interacting with Pg receptors. Certain Pg receptors also protect Pg activators and Pm from inactivation. Thus, Pg receptors such as AIIt are the key regulators of Pm generation.

Plasmin is crucial for dissolving fibrin clots in a process known as fibrinolysis. Fibrin blood clot formation begins when a break in a blood vessel allows plasma to contact tissue factors in the extracellular matrix. Initially, factor VII binds to tissue factor and is activated, which participates in the activation of a clotting factor cascade and results in the cleavage of prothrombin into thrombin. This thrombin activates platelets, resulting in the generation of FXa/Va complexes and the generation of sufficient thrombin from prothrombin to convert fibrinogen to fibrin, resulting in the formation of a stable hemostatic plug [84,85]. The fibrinolytic surveillance system is critical in preventing the inappropriate formation of blood clots and in regulating the size of blood clots that form at the site of injury. Fibrinolysis is initiated by the secretion of plasminogen activators (tPA and uPA). The endothelium of blood vessels secretes tPA, which circulates in the blood as a single-chain protein (sct-PA). The kidney secretes uPA, which circulates in an inactive single-chain form (scu-PA), which Pm must convert into the active two-chain form (tcu-PA) [86,87].

Pm degrades the fibrin blood clot, producing D-dimer and other fibrin degradation products. Impaired breakdown of fibrin clots is associated with many diseases, including diabetes, insulin resistance, sepsis, stroke, and the metabolic syndrome [88,89,90,91]. Impaired fibrinolysis also contributes to deep venous thrombosis and pulmonary embolism [92]. Impaired fibrinolysis is a feature of ischemic stroke and is present in both the acute and convalescent phases of the disease [93,94,95].

Our laboratory has utilized the S100A10-null mouse to elucidate the function of S100A10 in fibrinolysis in vivo [96]. We began these studies with human endothelial cells (TIME), in which S100A10 had been depleted by shRNA. We observed that these cells bound 50% less plasminogen and generated 60% less plasmin. We also isolated lung endothelial cells from WT and S100A10-null mice and observed that the endothelial cells from the S100A10-null mice bound 40% less plasminogen and generated 40% less plasmin. These studies suggested an important role for S100A10 in the fibrinolytic surveillance system. To test this possibility, we injected 125I-fibrinogen into the tail vein of WT and S100A10-null mice, followed by the injection of batroxobin, to initiate clot formation. We then isolated tissues, and observed that the tissues of the S100A10-null mice had a significantly greater accumulation of 125I-label than the WT mice. This suggested that S100A10-null mice had an inability to clear fibrin clots. Consistent with this study, when we examined the tissues of WT and S100A10-null mice’s tissues, we observed the presence of fibrin clots in the S100A10-null mice but not in the WT mice. This study showcased the importance of S100A10 in the endothelial cell fibrinolytic surveillance system in vivo and has broad-reaching implications for stroke, cardiovascular diseases, and cancer progression.

## 6. Neurological Functions

Previous reviews of S100A10 have uncovered new insight into the protein’s role in regulating mood-related behavior [97]. S100A10 has been implicated in the pathophysiology of depression and is being examined as a critical modulator of neurological functions [97]. Several G protein-coupled receptors, channels, and transporters have been identified to interact with S100A10, including serotonin 5-HT receptors [40], metabotropic glutamate receptor mGluR5 [98], Na^2+^ ion channel NaV1.8 [62], acid-sensing channel ASIC-1 [58], K^+^ channel subfamily K member 3 (TASK-1) [65,66], and transient receptor potential cation Ca^2+^ channels subfamily TRPV5 and TRPV6 [67]. Global proteomic screens have additionally identified C-C chemokine receptor type 10 (CCR10) [99], flottilin-1 (FLOT1) [44], follicle-stimulating hormone receptor (FSHR) [100], olfactory receptor 4N2 (OR4N2), and 4N1 (OR14l1) [43] as S100A10 interactors.

S100A10 expression is widespread in the brain and spans multiple structures and cell types. Milosevic et al. suggest that the expression is distinct to specific regions when comparing neuronal and nonneuronal cell types [101]. Some identified areas include the hippocampus [45,102], amygdala [102], cerebral cortex [103], anterior cingulate cortex [40], and nucleus accumbens [104]. The expression of S100A10 in the brain has been extensively examined for its role in modulating mood-related behaviors, including major depressive disorder, Parkinson’s disease, and other neuropsychiatric disorders [105,106].

The role of S100A10 in depression and as a regulator of the antidepressant response has been examined in various mouse and human studies. Reduced S100A10 expression in multiple brain regions in depressed individuals implicated S100A10′s role in depression pathology [40,102,107]. Epigenetic studies of S100A10 propose it as a biomarker to predict treatment response and diagnose depressive-like behavior. Expression of S100A10 can be regulated by various factors, such as antidepressants, nitric oxide, growth factors, and dexamethasone [108,109]. While S100A10 is reduced in brain tissue from depressed patients, S100A10 expression increases in the peripheral blood. One study comparing S100A10 expression levels by quantitative PCR found that, compared to healthy patients, depressed and high-stressed individuals studied had higher expression of S100A10 in their blood cells [110].

The S100A10-knock-out mouse models have been instrumental in studying the function of S100A10 in depression, mood disorders, and anxiety. In one of the first hallmark studies, Svenningson et al. showed that S100A10-depleted mice demonstrate depression-like behaviors, and S100A10 expression is increased in rodent brains after anti-depressant therapy. S100A10 mediates its function in depression via interaction with the serotonin 1B receptor [5-hydroxytrptamin (5HT_1B_) receptor] [40]. More recently, Seo et al. observed that S100A10 is a key protein that modulates chronic stress-induced depression in rodent models, and anti-depressants reverse this phenotype by upregulating S100A10 in the prelimbic cortex (PrL) [109]. Furthermore, they showed that S100A10 in ependymal cells regulates depressive states in chronic stress by modulating the flow of cerebrospinal fluid (CSF) by maintaining the planar cell polarity in ependymal cells [109].

## 7. Inflammation and Wound Healing

The plasminogen activation system and S100A10 play an integral role in the inflammatory process. S100A10 functions on the plasma membrane and in the extracellular space, correlated with the immune response and regulation of immune activities [111]. The Ca^2+^ binding activity of S100A proteins plays an important role during the inflammatory process by regulating different molecules and inflammatory signaling pathways that lead to inflammation [112,113]. S100 isoforms can contribute to the immune response as pro-inflammatory stimulators, chemo-attractants, and antimicrobial peptides [111].

During inflammation and wound healing processes, extracellular matrix (ECM) degradation is a vital step, and the S100A10 dimer component of Allt is a crucial molecule in the process [112]. Pg binds to the S100A10 dimer of Allt, and plasminogen activators tPA and uPA mediate its activation into Pm, which can initiate downstream proteolytic cascades associated with the wound repair process [32,34,60,114,115,116]. This S100A10-mediated protease activation is utilized by highly motile cells, such as metastatic cancer cells and macrophages, to facilitate its migration through the ECM [114,116,117,118,119,120,121,122,123].

Macrophages play a critical role in the pathogenic inflammatory response through initiating, maintaining, and resolving inflammation processes [119,124]. Cell-surface generation of plasmin is required for macrophage recruitment [125]. This recruitment is partly mediated through the plasmin-dependent activation of matrix metallopeptidase 9 (MMP-9) [119,125]. Activation of plasmin occurs through four plasminogen receptors found on the cell surface, including S100A10 [32,34,35,126], α-enolase [127], Plg-RKT [128], and histone H2B [129].

We have previously demonstrated a direct involvement of S100A10 in response to inflammatory stimuli to recruit macrophages [119]. P11 was up-regulated in macrophages that were activated by inflammatory mediators. Thioglycollate-stimulated peritoneal macrophages had higher S100A10 and annexin A2 protein levels when compared to resident peritoneal macrophages. In S100A10-/- mice, macrophage migration into the peritoneal cavity across the peritoneal membrane was decreased. S100A10 and the other carboxyl-terminal Pg receptors contribute to Pm generation in macrophages, which allows macrophages to play a role in directly facilitating the proteolysis of the basement membrane, hydrolyzing ECM proteins, and activating MMP-9 during inflammatory responses [119].

S100A10 can also activate human and murine macrophages directly through the toll-like receptor 4 (TLR4) pathway [130,131]. Cell surface Pm generated by Allt can trigger the phosphorylation of PKC signaling molecules, which can activate mitogen-activated protein kinase (MAPK), TLR4, and NFκB signaling pathways [130]. Allt disassembly after Allt phosphorylation can activate the CD11b-dependent integrin-linked kinase (ILK) pathway [132], which, together with Pm, can induce NFκB nuclear translocation and promote pro-inflammatory factor production [26,130,133]. The release of these pro-inflammatory cytokines, including IL-1, IL-6, and TNFα, can facilitate immune-escape mechanisms by evading immunosurveillance and mitigating T cell cytotoxicity [26,130,133]. Allt-driven cytokine production is inhibited by TLR-4 knockdown [130], which suggests that TLR-4 is important for Allt-mediated inflammation.

More recently, S100A10 has been examined for its role in the inflammatory condition of COVID-19 patients. The S100 family of proteins has been suggested to be able to direct more monocytes and neutrophils to the target site of COVID-19 patients by controlling the cytokine release syndrome [113]. A study comparing NK and NK-T cell subsets between COVID-19 patients and healthy individuals identified S100A10 as a marker for COVID-19-derived NK-T cells [134]. Disease trajectory transcriptomic models of COVID-19 severity from peripheral mononuclear cells found that genes encoding Ca^2+^-binding proteins play important roles in regulating inflammatory pathways; however, p11 was not indicated as one of the S100 proteins involved [135]. Another study looking at peripheral blood samples from COVID-19 cases observed an upregulation of S100A10, S100A4, and S100A9 mRNA [113].

## 8. Exocytosis and Trafficking

The AIIt complex has long been implicated at the plasma membrane to play a role in early and late secretory events and membrane-cytoskeleton lineage [47,136,137]. At the cell membrane, the AIIt complex was found localized to the contact site of the plasma membrane/secretory granule, suggesting it played a role in the exocytotic process of the cell [138,139,140,141]. While the crosslinking activity of the AIIt complex is necessary for efficient, regulated exocytosis, it is not an obligatory component [137]. The exocytotic machinery of the Allt complex is Ca^2+^ sensitive [142] and is more efficient than monomeric annexin protein in regulating Ca^2+^-dependent exocytosis [138,143,144]. These features and functions of the AIIt complex have played an important role in the pathogenesis of many viruses, such as HBV, HPV, and HIV-1 [145,146,147,148,149]. The AIIt complex has been shown to facilitate the exocytosis of von Willebrand factor (vWF) in vascular endothelial cells [150,151], the release of the bluetongue virus (BTV) in BHK21 cells [152,153], and the human papilloma virus type 16 in human keratinocytes [154]. Finally, recent studies by Bai et al. have shown that the AIIt complex promotes hepatitis B virus (HBV) exocytosis in the trophoblasts by recruiting VAMP2 and SNAP25 for membrane fusion events [145].

## 9. Autophagy and Metabolism

Cancer cells require increased metabolism to maintain sustained proliferative growth and spread. S100A10 can respond to a variety of signals to maintain and accelerate cancer cell metabolism. Members of the S100 family of proteins are involved in a number of metabolic functions, including redox, energy, and sugar metabolism [15]. S100A10 has been shown to increase the malignant growth of cancer cells by activating the mTOR signaling pathway in osteosarcoma [155], gastric cancer [156], pancreatic ductal adenocarcinoma (PDAC) [157,158], and HCC [159,160]. Li et al. reported that S100A10, through its interaction with annexin A2, accelerated tumor glycolysis and lactate production and contributed to the switch from oxidative phosphorylation to aerobic glycolysis [156]. Glucose consumption was significantly increased in S100A10-overespressing cells and reduced in S100A10-knockout cells. This overexpression of S100A10 also reduced the amount of intracellular ATP production, indicating the important role of S100A10 in facilitating glycolysis. By modulating the Src/annexin A2/AKT/mTOR signaling pathway, S100A10 could promote pro-tumor aerobic glycolysis, suppress cell apoptosis, and maintain cell proliferation.

S100A10 knockdown was also shown to affect the malignant growth of osteosarcoma cells by modulating glycolysis [155]. Knockdown of S100A10 in the osteosarcoma cell line inhibited proliferation, migration, and invasion and induced apoptosis via the AKT/mTOR pathway by modulating glycolysis. S100A10 played a critical role in HCC progression through the epithelial-mesenchymal transition by upregulating the epidermal growth factor receptor and AKT/ERK signaling pathways. Furthermore, Lin et al. showed that S100A10 could activate LAMB3 through the JNK pathway in PANC-1 cells [157]. By activating the JNK/LAMB3-LAMC2 axis, S100A10 was able to promote PDAC cell proliferation, migration, and adhesion.

## 10. Regulation of CFTR Function

The cystic fibrosis conductance regulator protein (CFTR) is a cAMP/protein kinase A (PKA) and ATP-regulated Cl^-^ channel. The mature, wild-type CFTR protein localizes to the apical membrane, where it controls fluid and ion transport. Mutations occurring in the Cl^-^ channel can manifest themselves in a number of disorders, most notably cystic fibrosis (CF). CFTR is surrounded by an interconnected, dynamic network of components that impact its location, trafficking, and functioning within the cell. Mutations in CFTR have previously been shown to disrupt the intracellular trafficking of the protein [161,162]. Defective endocytosis, intracellular protein trafficking, and exocytosis have also been observed in CF [161,162], suggesting that CFTR may interact with and modulate proteins within the secretory pathway.

We’ve previously discussed how AIIt has been implicated in endocytic and secretory events occurring close to or at the plasma membrane. Annexins regulate vesicular traffic [163] and are involved in the regulation of inflammation. A well-characterized annexin involved in the inflammatory response is annexin A1 [164,165]. Annexin A1 expression was down-regulated in cftr^-/-^ mice and CF patient epithelial samples, suggesting that in CF pathogenesis, the interaction of CFTR and annexin A1 may be a key process [166]. Annexins function at the plasma membrane and regulate vesicular traffic [163], and when compared to CFTR, they share significant sequence homology around the area of the most common CF mutation [167].

Annexins can interact with cytoskeletal proteins, such as ion channels, to modulate membrane events. CFTR is regulated by cAMP/PKA [168,169]. Borthwick et al. noted that cAMP/PKA could regulate Allt in cells, which led the group to speculate about the functional relationship among those different proteins [170]. They discovered that the calcineurin (CaN)-like phosphatase, mediated via cAMP/PKA, can dephosphorylate annexin A2, which results in it forming a complex with S100A10 [170]. Pre-treatment of the cells with a specific peptide corresponding to the annexin A2 binding site on S100A10 or with a CaN inhibitor before forskolin stimulation significantly reduced CFTR function in the cells. They concluded that the Allt/PKA/calcineurin/CFTR interacting complex is required for proper channel activity and tethering to the plasma membrane [170,171].

Compared to binding motifs on other ion channels, CFTR lacks a S100A10 binding motif [172]. The authors suggest that the cAMP and Ca^2+^-dependent pathways control the phosphorylation and dephosphorylation of annexin A2 within the Allt complex to regulate the interaction between Allt and CFTR. Another study suggests that the interaction between S100A10 and CFTR involves the NBD1 domain of CFTR [173]. S100A10 acts as an adapter, connecting the annexin A1/cytosolic phospholipase A2 (cPLA2) complex to CFTR. Annexin A1 failed to bind to the NBD1 region of CFTR, while S100A10 showed significant binding, indicating that in vitro, through NBD1/S100A10 binding, CFTR interacts with the cPLA2/annexin A1 complex to regulate inflammation. These studies provide evidence that S100A10 associates with CFTR, and this interaction plays a role in regulating ion homeostasis in epithelial cells. This research supports their hypothesis that AIIt affects CFTR function and provides additional evidence that its activation occurs through a cAMP/PKA-dependent process.

## 11. Mammalian Oviduct

S100 proteins that were associated with the mammalian oviduct were incidentally discovered by Nakamura et al., who reported S100 protein immunoreactivity in glandular cells of the cervix and placenta during early pregnancy [174]. S100 proteins were found to be linked to a differentiation phenomenon, and their expression was hypothesized to correlate with a defined phase of the cell cycle [175]. The presence of S100 in free-floating cells (differentiated) compared to epithelial cells (dedifferentiated cells) cultured from bovine oviducts further supported the hypothesis that S100 was associated with defined cell cycle phases [176]. When Nakamura et al. compared S100 protein immunoreactivity in non-pregnant women, the glandular cells of the cervix and endometrium showed no immunoreactivity to S100 protein staining. No immunoreactivity was observed in endometrial carcinomas or hyperplasia, suggesting a relationship between S100 proteins and early pregnancy. The number of S100-positive glands decreased by mid to late gestation, and in term placentas were almost absent, suggesting a relationship between the expression of S100 proteins and humoral factors related to pregnancy and the mammalian oviduct [174].

Reactivity for S100 proteins has been observed in all types of epithelial cells, segments of the fallopian tube [174,175,176], lymph vessels, blood capillaries, arterial vessels, and oocytes [177]. There was no change in reactivity observed during the oestrous cycle; however, S. Agarwal did observe a difference in reactivity between fallopian tubes that were removed for sterilization compared to ones removed for ectopic pregnancies, with the latter showing strong S100 reactivity [178].

The majority of studies examining S100A10 expression in the oviduct have been performed in non-human models. S100A10 is expressed at negligible levels in the mouse oviduct compared to S100A11, while both are found at substantial levels in the ovary [179]. PCR analysis of S100A10 in the mouse oviduct showed significantly lower expressions of S100A10 in the oviduct compared to the ovary, which suggests it is not the most prominent S100 protein in the oviduct. Similarly, Tingaud–Sequeira et al. reported significantly up-regulated S100A10 expression in the atretic ovaries and follicles of Senagalese sole fish [180]. The presence of S100A10 and annexin A2 has additionally been observed in the oviducts of rabbits, dogs, cats, cows, pigs, and humans [181].

## 12. Stemness and Cancer Pathogenesis

Tumors contain a subpopulation of cells, termed cancer stem cells (CSCs), that possess indefinite proliferative potential. CSCs possess embryonic characteristics and are drivers of tumor development [182,183]. Mechanistically, S100A10 has been examined to regulate expression levels of stem-cell-related genes. S100A10 was first suggested to confer stem-cell activity regulation by King et al., where they demonstrated that annexin A2 and S100A10 expression in Xenopus laevis larvae after limb amputation were up-regulated [184]. The expression of S100A10 was 4-fold higher in the regeneration-competent blastema compared to the time of amputation, in a pattern that was distinct from other immune-related genes [184]. This was the first evidence of S100A10 stemness-related activity.

CSCs have been examined for their role in tumor recurrence and metastasis. One of the reasons that chemotherapy resistance arises is the inability to eliminate the tumor stem cells completely. Alterations in mRNA and protein expression of the S100 protein family in various cancers participate in regulating drug resistance and conferring CSC properties [185]. S100A10 upregulation after chemotherapy treatment has been observed in colorectal cancer [186,187], leukemia [188], ovarian cancer [189,190], breast cancer [72,191], and neuroblastoma [192]. S100A10 can confer metastatic potential through mechanisms such as an upregulation of cellular invasion and CSC properties [191]. In addition to CSCs, S100A10 stimulation was also found to enhance bone marrow-derived stem cell osteogenesis [193].

S100A10 expression has been associated with the hallmarks of cancer in several cancers, such as the brain, breast, lung, colorectal, kidney, ovarian, acute lymphoblastic leukemia, and pancreatic [194]. Our laboratory has contributed to understanding the function of S100A10 in the pathogenesis of pro-myelocytic leukemia (PML), pancreatic cancer, and breast cancer [195,196]. Since S100A10 is a multi-functional protein, it plays both plasminogen-dependent and independent roles in the progression of cancer, depending on the cellular context. For example, S100A10 promotes hepatocellular carcinoma, gastric cancer, and osteosarcoma cell proliferation and suppresses apoptosis via enhanced aerobic glycolysis through mTOR signaling [155,156,159,197]. In acute promyelocytic leukemia (APL), we have shown that cell surface S100A10 expression in APL cells is responsible for the increased fibrinolytic events associated with APL patients [71]. Induction of PML-RAR-α oncoprotein resulted in increased S100A10 expression on the cell surface, with a subsequent increase in plasmin generation. In the MMTV-PyMT mouse mammary cancer model, loss of S100A10 resulted in dramatic impairment of tumor progression and pulmonary metastasis, but surprisingly, this was not mediated through loss of plasmin generation, as we did not observe a marked decrease in plasmin generation in the tumors lacking S100A10 [195]. Our previous studies using ectopic subcutaneous tumor models have suggested that plasmin generation and tumor cell infiltration by macrophages are substantially impaired in mice lacking S100A10, which substantially impedes tumor growth [119].

In hepatocellular carcinoma, S100A10 plays a pivotal role in tumor initiation, self-renewal capacity, chemoresistance, and metastasis, as validated in vivo and in experimental animal models. This study showed that S100A10 mediates these effects through two mechanisms. First, it enhanced metastasis by upregulating the expression of mesenchymal markers such as vimentin, fibronectin, and N-cadherin and promoting epithelial-mesenchymal transition. Second, the cancer cells also secreted S100A10 in the extracellular vesicles (EVs), which promoted motility, invasion, and metastasis. S100A10 potentially regulated the transfer of matrix metalloprotease 2 (MMP2), fibronectin, and epidermal growth factor (EGF) proteins into the EVs, thus indirectly regulating invasion and metastasis [159].

S100A10 is one of the 11 signature genes whose expression correlates with multidrug resistance in ovarian serous carcinoma [198]. Several other studies have also published similar observations in ovarian cancer, where increased S100A10 expression is associated with poor response to chemotherapy and overall survival [188,199]. The precise molecular mechanism by which S100A10 regulates drug resistance has yet to be discovered. One potential explanation is the role S100A10 plays in promoting cancer cell stemness, which has implications for promoting chemoresistance [72]. Nevertheless, elucidating the mechanism of S100A10 in therapy resistance is an interesting and necessary avenue for future research on S100A10.

**Table 3 biomolecules-13-01450-t003:** Suggested interactors of p11.

Interactor	Protein Name	Uniprot Accession	Cellular Location	Method	References
5-HTR_1B_	5-hydroxytryptamine receptor 1B	P28222	Plasma membrane	Two-hybrid	[40,41]
5-HTR_1D_	5-hydroxytryptamine receptor 1D	P28221	Plasma membrane	Two-hybrid	[41]
5-HTR4	5-hydroxytryptamine receptor 4	Q13639	Plasma membrane	Two-hybrid	[41]
ABCE1	ATP-binding cassette sub-family E member 1	P61221	Cytoplasm	AC-MS ^1^	[200]
AHNAK	Neuroblast differentiation-associated protein AHNAK	Q09666	Nucleus	AC-MS ^1^; AC-W ^2^; Co-crystal structure; Co-fractionation; PL-MS ^3^	[42,43,44,45]
ANG	Angiogenin	P03950	Extracellular; Nucleus	AC-W ^2^; Co-IP ^4^; Fluorescence imaging; Proximity ligation	[201]
ANLN	Anillin	Q9NQW6	Nucleus; Cytoplasm	AC-MS ^1^	[202]
ANTXR1	Anthrax toxin receptor 1	Q9H6X2	Plasma membrane	AC-MS ^1^	[203]
ANXA2	Annexin A2	P07355	Nucleus; Cytoplasm; Plasma membrane; Extracellular	AC-MS ^1^; AC-W ^2^; Co-crystal structure; Co-fractionation; PL-MS ^3^; Reconstituted complex; Two-hybrid	[43,45,46,47,48,49,50,51,52,53,54,55,56,57]
ASIC-1 channels	Acid-sensing ion channel 1	P78348	Plasma membrane	Two hybrid	[58]
ATF2	Cyclic AMP-dependent transcription factor ATF-2	P15336	Nucleus; Cytoplasm	AC-MS ^1^	[204]
ATP6V1E1	V-type proton ATPase subunit E	P36543	Cytoplasm; Plasma membrane	Co-fractionation	[44]
BAD	BCL2-associated agonist of cell death	Q92934	Cytoplasm	Reconstituted complex; Two-hybrid	[59]
BAP1	Ubiquitin carboxyl-terminal hydrolase BAP1	Q92560	Nucleus	AC-MS ^1^	[205]
BMI1	Polycomb complex protein BMI-1	P35226	Nucleus; Cytoplasm	Co-IP ^4^	[206]
Cathepsin B	Cathepsin B	P07858	Cytoplasm; Plasma membrane; Extracellular	Two hybrid; Reconstituted complex	[60]
CCDC171	Coiled-coil domain-containing protein 171	Q6TFL3	Nucleus	PL-MS ^3^	[43]
CCNF	Cyclin-F	P41002	Nucleus	AC-MS ^1^	[207]
CCR10	C-C chemokine receptor type 10	P46092	Plasma membrane	Proximity ligation; Pull-down	[99]
CD55	Complement decay-accelerating factor	P08174	Plasma membrane; Extracellular	Co-fractionation	[44]
CD81	CD81 antigen	P60033	Plasma membrane; Cytoplasm	Co-IP ^4^	[208]
CDC25C	M-phase inducer phosphatase 3	P30307	Nucleus; Cytoplasm	AC-MS ^1^	[209]
CDK16	Cyclin-dependent kinase 16	Q00536	Cytoplasm	Two-hybrid	[210]
CDK9	Cyclin-dependent kinase 9	P50750	Nucleus; Cytoplasm	AC-MS ^1^	[211]
CEP192	Centrosomal protein of 192 kDa	Q8TEP8	Cytoplasm	AC-MS ^1^	[212]
CFTR	Cystic fibrosis transmembrane conductance regulator	P13569	Plasma membrane; Cytoplasm	Surface plasmon resonance; Pull-down	[170,173,213]
CHMP4B	Charged multivesicular body protein 4b	Q9H444	Cytoplasm	AC-MS ^1^	[202]
CHMP4C	Charged multivesicular body protein 4c	Q96CF2	Cytoplasm	AC-MS ^1^	[202]
CIT	Citron Rho-interacting kinase	O14578	Cytoplasm	AC-MS ^1^	[202]
COPS6	COP9 signalosome complex subunit 6	Q7L5N1	Cytoplasm; Nucleus	Two-hybrid	[214]
Cytosolic Phospholipase A2		Q9UP65	Cytoplasm; Plasma membrane	Two-hybrid	[61]
DLC1	Rho GTPase-activating protein 7	Q96QB1	Cytoplasm	AC-W ^2^	[215]
DNAAF10	Dynein axonemal assembly factor 10	Q96MX6	Cytoplasm; Nucleus	Co-fractionation	[44]
DUSP19	Dual specificity protein phosphatase 19	Q8WTR2	Cytoplasm	AC-MS ^1^	[209]
DYNLT1	Dynein light chain Tctex-type 1	P63172	Cytoplasm	PL-MS ^3^	[216]
DYRK1A	Dual specificity tyrosine-phosphorylation-regulated kinase 1A	Q13627	Nucleus	AC-MS ^1^	[217]
ECT2	Protein ECT2	Q9H8V3	Nucleus; Cytoplasm	AC-MS ^1^	[202]
EFTUD2	116 kDa U5 small nuclear ribonucleoprotein component	Q15029	Nucleus	AC-MS ^1^	[218]
EGFR	Epidermal growth factor receptor	P00533	Plasma membrane; Cytoplasm	AC-MS ^1^	[219]
ELAVL1	ELAV-like protein	Q15717	Cytoplasm; Nucleus	Co-fractionation	[44]
ELMOD1	ELMO domain-containing protein 1	Q8N336	Nucleus	PL-MS ^3^	[43]
ERBB3	Receptor tyrosine-protein kinase erbB-3	P21860	Plasma membrane; Extracellular	Two-hybrid; AC-MS ^1^	[220,221]
ERBB4	Receptor tyrosine-protein kinase erbB-4	Q15303	Plasma membrane; Cytoplasm	Two-hybrid	[220]
ESR1	Estrogen receptor	P03372	Nucleus; Cytoplasm; Plasma membrane	AC-MS^1^	[222]
FAF2	FAS-associated factor 2	Q96CS3	Cytoplasm	Co-fractionation	[44]
FBL	rRNA 2′-O-methyltransferase fibrillarin	P22087	Nucleus	AC-MS ^1^	[212]
FLNA	Filamin-A	P21333	Cytoplasm	AC-MS ^1^	[50]
FLOT1	Flotillin-1	O75955	Plasma membrane	Co-fractionation	[44]
FSHR	Follicle-stimulating hormone receptor	P23945	Plasma membrane	Two-hybrid	[100]
GBP3	Guanylate-binding protein 3	Q9H0R5	Cytoplasm	PL-MS ^3^	[43]
GDNF	Glial cell line-derived neurotrophic factor	P39905	Extracellular	AC-MS ^1^	[51]
GOLGA4	Golgin subfamily A member 4	Q13439	Cytoplasm	PL-MS ^3^	[43]
HDAC4	Histone deacetylase 4	P56524	Cytoplasm; Nucleus	AC-MS ^1^	[223]
HDAC6	Histone deacetylase 6	Q9UBN7	Cytoplasm; Nucleus	AC-MS ^1^	[224]
HDLBP	Vigilin	Q00341	Cytoplasm; Nucleus	Co-fractionation; AC-MS ^1^	[44,225]
Heparin	Heparin cofactor 2	P05546	Cytoplasm; Extracellular	Pull-down	[226]
HLTF	Helicase-like transcription factor	Q14527	Nucleus; Cytoplasm	Co-fractionation; AC-MS ^1^; AC-W ^2^	[44,45]
HMGN3	High motility group nucleosome-binding domain-containing protein 3	Q15651	Nucleus	PL-MS ^3^	[43]
HSPA8	Heat shock cognate 71 kDa protein	P11142	Cytoplasm; Nucleus; Plasma membrane	AC-MS ^1^	[227]
KCNK3	Potassium channel subfamily K member 3	O14649	Plasma membrane	Reconstituted complex; Two-hybrid	[65]
KIAA1429	Protein virilizer homolog	Q69YN4	Nucleus; Cytoplasm	AC-MS ^1^	[228]
KIF14	Kinesin-like protein KIF14	Q15058	Cytoplasm; Nucleus	AC-MS ^1^	[202]
KIF20A	Kinesin-like protein KIF20A	O95235	Cytoplasm; Nucleus	AC-MS ^1^	[202]
LIMA1	LIM domain and actin-binding protein 1	Q9UHB6	Cytoplasm	AC-MS ^1^	[50]
LMX1B	LIM homeobox transcription factor 1β	O60663	Nucleus	AC-MS ^1^	[53]
LUC7L3	Luc7-like protein 3	O95232	Nucleus	AC-MS ^1^	[43]
MAEA	E3 ubiquitin-protein transferase MAEA	Q7L5Y9	Cytoplasm; Nucleus	PL-MS ^3^	[43]
MAPT	Microtubule-associated protein tau	P10636	Cytoplasm	Two-hybrid	[229]
mGluR5	Metabotropic glutamate receptor 5	P41594	Plasma membrane	AC-W ^2^	[98]
MLKL	Mixed lineage kinase domain-like protein	Q8NB16	Plasma membrane; Cytoplasm	Two-hybrid	[230]
MMGT1	ER membrane protein complex subunit 5	Q8N4V1	Cytoplasm	AC-MS ^1^	[231]
MYC	MYC proto-oncogene protein	P01106	Nucleus	AC-MS ^1^	[232]
MYH9	Myosin-9	P35579	Cytoplasm	AC-MS ^1^	[50]
Myo1c	Unconventional myosin-1c	O00159	Cytoplasm; Nucleus	AC-MS ^1^	[50]
NANOS2	Nanos homolog 2	P60321	Cytoplasm	Two-hybrid	[230]
NARS1	Asparagine—tRNA ligase, cytoplasmic	O43776	Cytoplasm	PL-MS ^3^	[43]
NaV1.8	Sodium channel protein type 8 subunit α	Q9UQD0	Plasma membrane	Two-hybrid	[62]
NBR1	Next to BRCA1 gene protein	Q14596	Cytoplasm	PL-MS ^3^	[233]
NEB	Nebulin	P20929	Cytoplasm	PL-MS ^3^	[43]
NPM1	Nucleophosmin	P06748	Nucleus; Cytoplasm	AC-MS ^1^	[234]
NUDCD1	NudC domain-containing protein 1	Q96RS6	Nucleus; Cytoplasm	Co-IP ^4^	[235]
OR14I1	Olfactory receptor 14I1	A6ND48	Plasma membrane	PL-MS ^3^	[43]
OR4N2	Olfactory receptor 4N2	Q8NGD1	Plasma membrane	PL-MS ^3^	[43]
PCTAIRE-1	Cyclin-dependent kinase 16	Q00536	Cytoplasm	Two-hybrid	[63]
PDYN	Proenkephalin-B	P01213	Extracellular	AC-MS ^1^	[50]
PHF5A	PHD finger-like domain-containing protein 5A	Q7RTV0	Nucleus	Co-fractionation	[44]
PIK3R6	Phosphoinositide 3-kinase regulatory subunit 6	Q5UE93	Cytoplasm; Plasma membrane	PL-MS ^3^	[53]
PLA2G4A	Cytosolic phospholipase A2	P47712	Cytoplasm; Nucleus	AC-W ^2^; Reconstituted complex	[46,55,236]
PLA2G4C	Cytosolic phospholipase A2γ	Q9UP65	Cytoplasm	Reconstituted complex	[61]
PLA2R	Urokinase plasminogen activator surface receptor	Q03405	Plasma membrane	AC-MS ^1^	[237]
PLD1	Phospholipase D1	Q13393	Cytoplasm	PL-MS ^3^	[43]
Plg	Plasminogen	P00747	Extracellular	AC-W ^2^; Surface plasmon resonance	[32,33]
PPFIA3	Liprin-α-3	O75145	Plasma membrane; Cytoplasm	PL-MS ^3^	[43]
PPIF	Peptidyl-prolyl cis-trans isomerase F	P30405	Cytoplasm	Co-fractionation	[238]
PRC1	Protein regulator of cytokinesis 1	O43663	Nucleus; Cytoplasm	AC-MS ^1^	[202]
RAF1	RAF proto-oncogene serine/threonine protein kinase	P04049	Cytoplasm; Plasma membrane; Cytoplasm	AC-MS ^1^	[225]
RPL10A	60S ribosomal protein L10a	P62906	Cytoplasm	Co-fractionation	[44]
RPL10L	60S ribosomal protein L10-like	Q96L21	Cytoplasm	Co-fractionation	[44]
RPL12	60S ribosomal protein L12	P30050	Nucleus; Cytoplasm	Co-fractionation	[44]
RYBP	RING1 and YY1-binring protein	Q8N488	Nucleus; Cytoplasm	Co-IP ^4^	[206]
S100A10	Protein S100-A10	P60903	Plasma membrane; Cytoplasm; Nucleus; Extracellular	AC-MS ^1^; Co-crystal structure	[45,54]
S100A3	Protein S100-A3	P33764	Cytoplasm	Two-hybrid	[230]
S100A7	Protein S100-A7	P31151	Cytoplasm; Extracellular	AC-MS ^1^	[64]
S100A8	Protein S100-A8	P05109	Plasma membrane; Cytoplasm; Extracellular	AC-MS ^1^	[64]
S100Z	Protein S100-Z	Q8WXG8	Cytoplasm	Two-hybrid	[230]
SETDB1	Histone-lysine N-methyltransferase SETDB1	Q15047	Nucleus	Two-hybrid	[214]
SIN3A	Paired amphipathic helix protein Sin3A	Q96ST3	Nucleus; Cytoplasm	Co-fractionation	[44]
SLC25A46	Mitochondrial outer membrane protein	Q96AG3	Cytoplasm	PL-MS ^3^	[43]
SLC8A1	Sodium/calcium exchanger 1	P32418	Plasma membrane	Florescence polarization spectroscopy	[239]
SLFN14	Protein SLFN14	P0C7P3	Nucleus	PL-MS ^3^	[43]
Smarca3	Helicase-like transcription factor	Q14527	Nucleus; Cytoplasm	Co-crystal structure	[45]
SRGAP3	SLIT-ROBO Rho GTPase-activating protein 3	O43295	Cytoplasm	PL-MS ^3^	[43]
SRP9	Signal recognition particle 9 kDa protein	P49458	Cytoplasm	Co-fractionation	[44]
SRPRB	Signal recognition particle receptor subunit β	Q9Y5M8	Cytoplasm	Co-fractionation	[44]
SUMO2	Small ubiquitin-related modifier 2	P61956	Nucleus; Cytoplasm	Reconstituted complex	[240]
SUPT6H	Transcription elongation factor SPT6	Q7KZ85	Nucleus; Cytoplasm	AC-MS ^1^; AC-W ^2^	[45]
SWI5	DNA repair protein SWI homolog	Q1ZZU3	Nucleus	PL-MS ^3^	[43]
TAB1	TGF-β-activated kinase 1 and MAP3K7-binding protein 1	Q15750	Cytoplasm	AC-MS ^1^	[53]
TASK-1	Potassium channel subfamily K member 3	O14649	Plasma membrane	Two-hybrid	[65,66]
TCEAL5	Transcription elongation factor A protein-like 5	Q5H9L2	Nucleus	AC-MS ^1^	[53]
TME65	Transmembrane protein 65	Q6PI78	Plasma membrane	Co-fractionation	[44]
TNF	Tumor necrosis factor	P01375	Plasma membrane; Extracellular	AC-MS ^1^	[241]
TNFRSF10A	Tumor necrosis factor receptor superfamily member 10A	O00220	Plasma membrane	AC-MS^1^; Co-IP ^4^	[52]
TP53	Cellular tumor antigen p53	P04637	Nucleus; Cytoplasm	Florescence polarization spectroscopy	[239]
Transglutaminase	Protein-glutamine-γ-glutamyltransferase K & 2	P22735; P08587	Plasma membrane; Cytoplasm	AC-W ^2^	[68]
TRIM37	E3 ubiquitin-protein ligase TRIM37	O94972	Cytoplasm	AC-MS ^1^	[242]
TRIM67	Tripartite motif-containing protein 67	Q6ZTA4	Cytoplasm	AC-MS ^1^	[243]
TRPM4	Transient receptor potential cation channel subfamily M member 4	Q8TD43	Plasma membrane	Florescence polarization spectroscopy	[239]
TRPV5	Transient receptor potential cation channel subfamily V member 5	Q9NQA5	Plasma membrane	Reconstituted complex; Two-hybrid	[67]
TRPV6	Transient receptor potential cation channel subfamily V member 6	Q9H1D0	Plasma membrane	Reconstituted complex	[67]
TTLL13	Tubulin polyglutamylase TTLL13	A6NNM8	Cytoplasm	PL-MS ^3^	[43]
UBAP2	Ubiquitin-associated protein 2	Q5T6F2	Cytoplasm; Nucleus	Co-fractionation	[44]
ULK1	Serine/threonine-protein kinase ULK1	O75385	Cytoplasm	Co-IP ^4^	[244]
UQCRB	Cytochrome b-c1 complex subunit 7	P14927	Cytoplasm	AC-MS ^1^	[52,53]
USP2	Ubiquitin carboxyl-terminal hydrolase 2	O75604	Cytoplasm	Two-hybrid	[230]
VIRMA	Protein virilizer homolog	Q69YN4	Nucleus; Cytoplasm	AC-MS ^1^	[228]
WDR92	Dynein axonemal assembly factor 10	Q96MX6	Nucleus	Co-fractionation	[44]
ZCCHC9	Zinc finger CCHC domain-containing protein 9	Q8N567	Nucleus	AC-MS ^1^	[53]
ZFR	Zinc finger RNA-binding protein	Q96KR1	Nucleus	Co-fractionation	[44]
ZGPAT	Zinc finger CCCH-type with G patch domain-containing protein	Q8N5A5	Nucleus	Two-hybrid	[230]
ZMPSTE24	CAAX prenyl protease 1 homolog	O75844	Cytoplasm	Co-fractionation	[44]
ZNF428	Zinc finger protein 428	Q96B54	Nucleus	AC-MS ^1^	[53]
ZRANB1	Ubiquitin thioesterase ZRANB1	Q9UGI0	Cytoplasm; Nucleus	AC-MS ^1^	[245]

^1^—Affinity Capture—Mass Spectrometry, ^2^—Affinity Capture—Western Blot, ^3^—Proximity Label—Mass Spectrometry, ^4^—Anti bait Co-Immunoprecipitation.

Depletion of S100A10 in pancreatic cancer cell lines resulted in decreased cell surface plasminogen activation and an overall reduction in cell invasiveness and tumor growth [157,196]. Bydoun et al. established that S100A10 expression was driven by promoter methylation and oncogenic RAS in pancreatic cancer. Recent studies by Lin et al. have suggested that S100A10 positively modulates pancreatic cancer cell proliferation, migration, adhesion, and in vivo tumor growth by activating laminin subunit β3 (LAMB3) via the JNK pathway. Thus, two different mechanisms are activated by S100A10 in pancreatic cancer, with the overall goal of accelerating tumor progression.

## 13. Summary and Conclusions

This review highlights the recent advances in the structure, regulation, binding partners, interactions, and functions of S100A10. As a member of the S100 family of Ca^2+^-binding proteins, S100A10 is unique in its permanent active conformation. Our work has highlighted just a few of the many important cellular functions of S100A10. Other laboratories have extended our studies and also added to the repertoire of additional regulatory functions of S100A10. The most noteworthy of these studies is the role of S100A10 as a prognostic biomarker in cancer and depression. Especially interesting is the revelation that S100A10 mRNA levels may predict the severity of COVID-19 infection. On a final note, one of the most unexplored potential functions of S100A10 is its role in the brain fibrinolytic surveillance system. This has major implications in terms of a physiological role for S100A10 in mitigating the deleterious effects of stroke and as a potential therapeutic agent for augmenting tPA-dependent thrombolytic therapy.

## Figures and Tables

**Figure 1 biomolecules-13-01450-f001:**
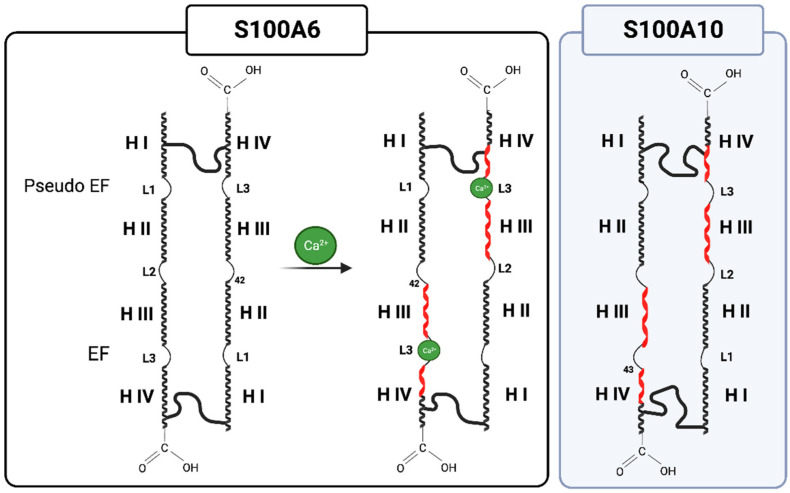
S100 (ex. S100A6) monomers, typically containing two EF-hand motifs consisting of a short helix, loop, and long helix, undergo a conformation change upon Ca^2+^ binding, exposing a hydrophobic patch in helix III and IV to bind target proteins (red helix). S100A10 resembles the Ca^2+^-loaded S100 protein, suggesting that it exists in an active, Ca^2+^-independent conformation. Created with “Biorender.com (accessed on 5 September 2023)”.

**Figure 2 biomolecules-13-01450-f002:**
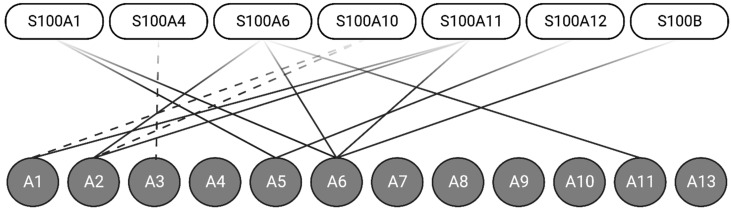
Seven S100 proteins can form Ca^2+^-dependent (solid-line) or Ca^2+^-independent (dashed-line) interactions with one of the twelve annexin proteins. Annexins A1, A2, A4, and A6 can form interactions with multiple S100 proteins. Created with “Biorender.com (accessed on 8 September 2023)”.

**Figure 4 biomolecules-13-01450-f004:**
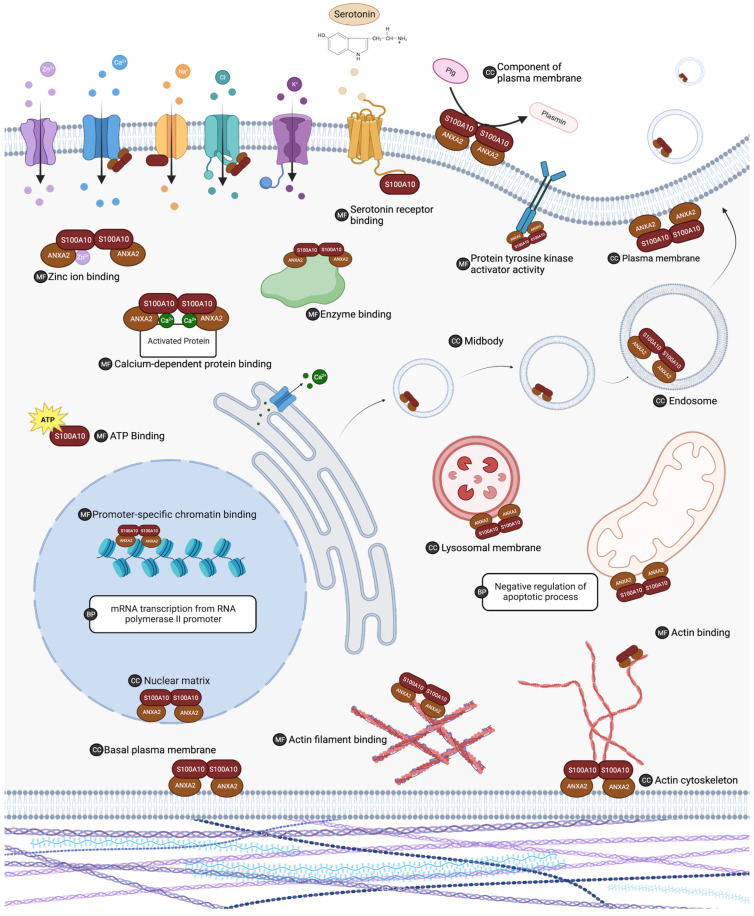
The list of S100A10 interacting proteins was subjected to gene ontology analysis in DAVID 7.0 to characterize putative biological activity. Gene ontology analysis indicated that S100A10 interacting proteins are ubiquitous in the cell and are involved in many important processes. Biological processes (BP), molecular functions (MF), and cellular component (CC) ontology results with a *p* value < 0.05 and FDR < 0.1 are summarized. ANXA2 (annexin A2). Created with “Biorender.com (accessed on 6 August 2023)”.

**Table 1 biomolecules-13-01450-t001:** Binding partners participating in different S100A10-related functions.

GO Classification	Ontology Terms	Proteins
Biological process	Negative regulation of apoptotic process (GO:0043066)	EGFR, ERBB3, ERBB4, FLNA, GDNF, KIF14, MYC, NPM1, PLA2R, PPIF, RAF1, SIN3A, TNF, TP53
Biological Process	mRNA transcription from RNA polymerase II promoter (GO:0042789)	ANXA2, ATF2, HLTF, S100A10, SUPT6H
Cellular component	Plasma membrane region (GO:0098590)	EGFR, ERBB3, ERBB4, PLD1
Cellular component	Midbody (GO:0030496)	ANLN, ANXA2, CHMP4B, CHMP4C, CIT, ECT2, KIF14, KIF20A, PRC1
Cellular component	Plasma membrane (GO:0005886)	5-HTR_1B_, 5-HTR_1D_, 5-HTR_4_, AHNAK, ANTXR1, ANXA2, ASIC-1 channels, CCR10, CD55, CD81, CDK16, CFTR, CHMP4B, CHMP4C, EGFR, ERBB3, ERBB4, ESR1, FLNA, FLOT1, FSHR, GOLGA4, HDLBP, HSPA8, KCNK3, KIF14, LIMA1, MAEA, MAPT, mGluR5, MLKL, MMGT1, MYH9, Myo1c, NaV1.8 sodium channel, OR14I1, OR4N2, PDYN, PIK3R6, PLA2G4C, PLA2R, PLD1, PLG, PRC1, RAF1, S100A10, S100A3, S100A8, SLC8A1, TMEM65, TNF, TNFRSF10A, Transglutaminase, TRPM4, TRPV5, TRPV6, ZGPAT
Cellular component	Basal plasma membrane (GO:0009925)	CD81, EGFR, ERBB3, ERBB4, Myo1c
Cellular component	Nuclear matrix (GO:0016363)	ANXA2, HLTF, MAEA, PHF5A, S100A10, TP53
Cellular component	Actin cytoskeleton (GO:0015629)	AHNAK, ANG, ANLN, FLNA, LIMA1, MYH9, Myo1c, NEB
Cellular component	Integral component of plasma membrane (GO:0005887)	5-HTR_1B_, 5-HTR_1D_, 5-HTR_4_, ASIC-1 channels, CCR10, CD81, CFTR, EGFR, ERBB3, ERBB4, FSHR, KCNK3, MAEA, mGluR5, PLA2R, SLC8A1, TNF, TRPM4, TRPV5, TRPV6
Cellular component	Lysosomal membrane (GO:0005765)	AHNAK, ANXA2, ATP6V1E1, CFTR, CHMP4B, CHMP4C, FLOT1, HSPA8, PLD1
Cellular component	Endosome (GO:0005768)	5-HTR4, ANXA2, ATP6V1E1, CHMP4B, EGFR, FLOT1, FSHR, PLD1
Molecular function	Calcium-dependent protein binding (GO:0048306)	ANXA2, S100A10, S100A3, S100A7, S100A8, S100Z
Molecular function	Zinc ion binding (GO:0008270)	BMI1, ESR1, HDAC4, HDAC6, HLTF, NANOS2, NBR1, PHF5A, Q6ZTA4, Q96KR1, S100A3, S100A7, S100A8, SETDB1, TP53, TRIM37, ZCCHC9
Molecular function	Promoter-specific chromatin binding (GO:1990841)	ATF2, BMI1, HDAC4, SETDB1, TP53
Molecular function	ATP binding (GO:0005524)	HSPA8, KIF14, KIF20A, MLKL, MYH9, Myo1c, NARS1, NaV1.8 sodium channel, RAF1, SLFN14
Molecular function	Enzyme binding (GO:0019899)	CFTR, EGFR, ESR1, HDAC6, HSPA8, MAPT, PLA2R, PLG, RAF1, TP53
Molecular function	Actin binding (GO:0003779)	ANG, ANLN, DYRK1A, HDAC6, MAEA, MAPT, MYH9, Myo1c, NEB
Molecular function	Calmodulin binding (GO:0005516)	ESR1, MYH9, Myo1c, SLC8A1, TRPM4, TRPV5, TRPV6
Molecular function	Serotonin binding (GO:0051378)	5-HTR_1B_, 5-HTR_1D_, 5-HTR_4_

**Table 2 biomolecules-13-01450-t002:** Proteins with known functional interactions with p11.

Interactor	Protein Name	Uniprot Accession	Cellular Location	Method	References
5-HTR_1B_	5-hydroxytryptamine receptor 1B	P28222	Plasma membrane	Two-hybrid	[40,41]
5-HTR_1D_	5-hydroxytryptamine receptor 1D	P28221	Plasma membrane	Two-hybrid	[41]
5-HTR4	5-hydroxytryptamine receptor 4	Q13639	Plasma membrane	Two-hybrid	[41]
AHNAK	Neuroblast differentiation-associated protein AHNAK	Q09666	Nucleus	AC-MS ^1^; AC-W ^2^; Co-crystal structure; Co-fractionation; PL-MS ^3^	[42,43,44,45]
ANXA2	Annexin A2	P07355	Nucleus; Cytoplasm; Plasma membrane; Extracellular	AC-MS ^1^; AC-W ^2^; Co-crystal structure; Co-fractionation; PL-MS ^3^; Reconstituted complex; Two-hybrid	[43,45,46,47,48,49,50,51,52,53,54,55,56,57]
ASIC-1 channels	Acid-sensing ion channel 1	P78348	Plasma membrane	Two hybrid	[58]
BAD	BCL2-associated agonist of cell death	Q92934	Cytoplasm	Reconstituted complex; Two-hybrid	[59]
Cathepsin B	Cathepsin B	P07858	Cytoplasm; Plasma membrane; Extracellular	Two hybrid; Reconstituted complex	[60]
Cytosolic Phospholipase A2		Q9UP65	Cytoplasm; Plasma membrane	Two-hybrid	[61]
NaV1.8	Sodium channel protein type 8 subunit α	Q9UQD0	Plasma membrane	Two-hybrid	[62]
PCTAIRE-1	Cyclin-dependent kinase 16	Q00536	Cytoplasm	Two-hybrid	[63]
Plg	Plasminogen	P00747	Extracellular	AC-W ^2^; Surface plasmon resonance	[32,33]
S100A7	Protein S100-A7	P31151	Cytoplasm; Extracellular	AC-MS ^1^	[64]
S100A8	Protein S100-A8	P05109	Plasma membrane; Cytoplasm; Extracellular	AC-MS ^1^	[64]
TASK-1	Potassium channel subfamily K member 3	O14649	Plasma membrane	Two-hybrid	[65,66]
TRPV5	Transient receptor potential cation channel subfamily V member 5	Q9NQA5	Plasma membrane	Reconstituted complex; Two-hybrid	[67]
TRPV6	Transient receptor potential cation channel subfamily V member 6	Q9H1D0	Plasma membrane	Reconstituted complex	[67]
Transglutaminase	Protein-glutamine-γ-glutamyltransferase K & 2.	P22735; P08587	Plasma membrane; Cytoplasm	AC-W ^2^	[68]

^1^—Affinity Capture—Mass Spectrometry, ^2^—Affinity Capture—Western Blot, ^3^—Proximity Label—Mass Spectrometry.

## Data Availability

The data presented in this study are openly available at NCBI “https://www.ncbi.nlm.nih.gov/ (accessed on 3 July 2023)”, BioGRID “https://thebiogrid.org/ (accessed on 4 July 2023)”, and ELIXIR “https://www.ebi.ac.uk/intact/home (accessed on 5 July 2023)”.

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
