# Peer review of "Recent Advances in Molecular and Cellular Functions of S100A10"

_biomolecules, 2023, doi:10.3390/biom13101450_

Round 1

Reviewer 1 Report

In this manuscript, Okura et al. provide a comprehensive review on the biological functions and binding partners of S100A10, an atypical member of the S100 family of calcium-binding proteins since it does not bind calcium. Indeed, in contrast to all other S100 proteins, which get activated through conformational changes upon calcium binding, S100A10 is always in an active, Ca-bound-like conformation for target binding. In vivo, however, due to its short lifetime as an isolated protein, S100A10 is most often associated in a complex with annexin A2, a complex whose activity is regulated by divalent cations. The authors of this review first give a structural overview of the S100 protein family, their calcium binding properties and their tendency to form complexes with annexins, with a strong focus on the annexin A2:S100A10 complex. Then they provide a rather exhaustive description of the biological processes in which S100A10 takes part, mostly as a complex with annexin A2, highlighting the binding partners and signaling pathways involved when known, as well as the pathophysiological consequences of these interactions.

Although many excellent reviews have already been published on S100A10, including some by the authors of the current manuscript, they most often dealt with one specific aspect of the function of this protein. The novelty of this manuscript is that it gives a complete  overview of the various fields in which S100A10 has an action, from neurological functions to inflammation, exocytosis, autophagy or metabolism, as well as in pathologies like cancers, cystic fibrosis or even COVID-19. The manuscript is well-written and clear for most part. My sole major concern is that it lacks illustrations since there is only one figure, and a long table, for the whole manuscript. I therefore have a few recommendations to improve the manuscript that I believe should be addressed before the manuscript can be considered for publication:

1) Lines 54-67: although the conformational changes occurring in EF-hand Ca-binding proteins upon Ca binding have been extensively described in other articles and reviews, and as it is an important point here to understand the structure-function relationship of S100A10, I strongly recommend that the authors add a figure describing these conformational changes at the structural level in both the S100 and calmodulin EF-hand families. This will help the reading, especially for non-structuralist readers.

2) Lines 63-67: the authors explain how S100 binds through their hydrophobic patches to their targets via their centrosymmetric dimers. But this is not necessarily true for all interactions of all S100 proteins:

- This sentence should be modulated by adding for example “in most cases”, to highlight that this statement may not be true for all S100-target complexes. For example, another dimeric organization has been observed structurally for S100A6 in complex with the RAGE receptor, where the two symmetrical binding sites for RAGE end up being on the same face.

- The symmetrical binding mode of S100 proteins could also be illustrated by a little scheme, on the same figure as the conformational changes occurring upon Ca-binding.

-  References are missing in that part

3) Lines 82-92: again, this part should be illustrated by a figure, showing for example the conformation of apo-S100A10 and how it matches well with that of other Ca-bound S100, with a closer view on the EF-hands.

4) All the part describing structural information on annexins and the S100:annexin complexes could also be illustrated with figures. Following the description in the text is not so obvious for readers non familiar with a structural point of view. This review would really gain in having more figures.

5) I would separate the paragraphs on the binding of S100A10 to annexin A2 from the part on general considerations regarding the structure of S100 proteins (i.e. Part 1 Structure from lines 28 to 92, and then Part 2 S100-annexin interactions).

6) The paragraph on the role of the ANX2/S100A10 complex with respect to CFTR function is not very clear. Has a direct interaction been proposed between ANX2/S100A10 and the CFTR channel? What is the outcome of this interaction on CFTR function? And how is it affected in a CF context, i.e. with a mutated, dysfunctional CFTR protein? I would recommend that the authors reformulate this part to make the answer to these questions, and thereby the role of ANX2/S100A10 in CFTR function, clearer.

7) While I appreciate that Figure 1 aims to give a general overview of all S100A10-related functions through interaction with different types of binding partners, since these binding partners are not properly named on the figure, it is rather difficult to follow. I think the specific binding partners and/or families of binding partners depicted in Figure 1, with obviously very specific topologies, should be named clearly so that one can really understand which processes are referred to in each part of the Figure. Alternatively, it could make more sense to make several figures to highlight specific binding partners and the signaling outcome of their interaction with S100A10, in each part of the review.

Minor details:

- Lines 16-17: “is located on chromosome 1q21”. The authors should specify that this location is for the mammalian proteins. Putative orthologs of S100A10 have been described in non-mammalian vertebrates but they are located at a distinct chromosomal position.

- Lines 71-74: the list of S100 proteins only holds true for mammals. There are 10 more orthologs discovered nowadays that are specific to certain non-mammalian families of vertebrates. So the authors should either include these in the list or specify that they only talk for mammals or even humans.

- Line 168: “Ca-binding affinity most of the S100 proteins”. I believe a “of” is missing after the word “affinity”.

- Line 170: the second “its” is confusing as one could believe it refers to S100A10 and not the target. The authors should clarify.

- Line 173: “S100A110”. Correct the typo.

- Line 417: “CF epithelial”. Word missing? Or epithelia instead of epithelial?

Author Response

Review 1.

In this manuscript, Okura et al. provide a comprehensive review on the biological functions and binding partners of S100A10, an atypical member of the S100 family of calcium-binding proteins since it does not bind calcium. Indeed, in contrast to all other S100 proteins, which get activated through conformational changes upon calcium binding, S100A10 is always in an active, Ca-bound-like conformation for target binding. In vivo, however, due to its short lifetime as an isolated protein, S100A10 is most often associated in a complex with annexin A2, a complex whose activity is regulated by divalent cations. The authors of this review first give a structural overview of the S100 protein family, their calcium binding properties, and their tendency to form complexes with annexins, with a strong focus on the annexin A2:S100A10 complex. Then they provide a rather exhaustive description of the biological processes in which S100A10 takes part, mostly as a complex with annexin A2, highlighting the binding partners and signaling pathways involved when known, as well as the pathophysiological consequences of these interactions.

--Authors comments–Thank you for your kind comments and for providing a meticulous review of the manuscript.

Although many excellent reviews have already been published on S100A10, including some by the authors of the current manuscript, they most often dealt with one specific aspect of the function of this protein. The novelty of this manuscript is that it gives a complete  overview of the various fields in which S100A10 has an action, from neurological functions to inflammation, exocytosis, autophagy, or metabolism, as well as in pathologies like cancers, cystic fibrosis or even COVID-19. The manuscript is well-written and clear for most part. My sole major concern is that it lacks illustrations since there is only one figure, and a long table, for the whole manuscript. I therefore have a few recommendations to improve the manuscript that I believe should be addressed before the manuscript can be considered for publication:

1) Lines 54-67: although the conformational changes occurring in EF-hand Ca-binding proteins upon Ca binding have been extensively described in other articles and reviews, and as it is an important point here to understand the structure-function relationship of S100A10, I strongly recommend that the authors add a figure describing these conformational changes at the structural level in both the S100 and calmodulin EF-hand families. This will help the reading, especially for non-structuralist readers.

--We have added a diagram. 

2) Lines 63-67: the authors explain how S100 binds through their hydrophobic patches to their targets via their centrosymmetric dimers. But this is not necessarily true for all interactions of all S100 proteins:

This sentence should be modulated by adding for example “in most cases”, to highlight that this statement may not be true for all S100-target complexes. For example, another dimeric organization has been observed structurally for S100A6 in complex with the RAGE receptor, where the two symmetrical binding sites for RAGE end up being on the same face.

--Agreed–we have added “in most cases”. We also added “In contrast, a unique dimeric structural organization has been observed for the S100A6/ RAGE receptor complex, where the two symmetrical binding sites for the RAGE receptor are localized to the same face of the S100A6 dimer (13).

The symmetrical binding mode of S100 proteins could also be illustrated by a little scheme, on the same figure as the conformational changes occurring upon Ca-binding.

References are missing in that part

--This has been corrected.

3) Lines 82-92: again, this part should be illustrated by a figure, showing for example the conformation of apo-S100A10 and how it matches well with that of other Ca-bound S100, with a closer view on the EF-hands.

--A figure has been added.

4) All the part describing structural information on annexins and the S100:annexin complexes could also be illustrated with figures. Following the description in the text is not so obvious for readers non familiar with a structural point of view. This review would really gain in having more figures.

--We have added the requested figures.

5) I would separate the paragraphs on the binding of S100A10 to annexin A2 from the part on general considerations regarding the structure of S100 proteins (i.e. Part 1 Structure from lines 28 to 92, and then Part 2 S100-annexin interactions).

--The paragraph has been separated so that we are talking about the binding of S100A10 to annexin A2 in one section. Part 1 on the structure of S100 proteins is now from lines 56 to 105. The binding of S100A10 to annexin A2 can now be found from 125 to 163. 

6) The paragraph on the role of the ANX2/S100A10 complex with respect to CFTR function is not very clear. Has a direct interaction been proposed between ANX2/S100A10 and the CFTR channel? What is the outcome of this interaction on CFTR function? And how is it affected in a CF context, i.e. with a mutated, dysfunctional CFTR protein? I would recommend that the authors reformulate this part to make the answer to these questions, and thereby the role of ANX2/S100A10 in CFTR function, clearer.

--This paragraph has been re-written to make the role of ANXA2/S100A10 in the context of CFTR function more clear. We have added a reference that discusses the potential interactions proposed between the two complexes. 

7) While I appreciate that Figure 1 aims to give a general overview of all S100A10-related functions through interaction with different types of binding partners, since these binding partners are not properly named on the figure, it is rather difficult to follow. I think the specific binding partners and/or families of binding partners depicted in Figure 1, with obviously very specific topologies, should be named clearly so that one can really understand which processes are referred to in each part of the Figure. Alternatively, it could make more sense to make several figures to highlight specific binding partners and the signaling outcome of their interaction with S100A10, in each part of the review.

-- We have added a table with the specific family of binding partners to clarify what proteins were used to determine the S100A10-related functions. 

Minor details:

-- Corrected all.

- Lines 16-17: “is located on chromosome 1q21”. The authors should specify that this location is for the mammalian proteins. Putative orthologs of S100A10 have been described in non-mammalian vertebrates but they are located at a distinct chromosomal position.

- Lines 71-74: the list of S100 proteins only holds true for mammals. There are 10 more orthologs discovered nowadays that are specific to certain non-mammalian families of vertebrates. So the authors should either include these in the list or specify that they only talk for mammals or even humans.

- Line 168: “Ca-binding affinity most of the S100 proteins”. I believe a “of” is missing after the word “affinity”.

- Line 170: the second “its” is confusing as one could believe it refers to S100A10 and not the target. The authors should clarify.

- Line 173: “S100A110”. Correct the typo.

- Line 417: “CF epithelial”. Word missing? Or epithelia instead of epithelial?

Reviewer 2 Report

The manuscript by Okura et al reviews the functions and the binding partners of S100A10. The table 1 provides a comprehensive summary from the literature of all known ligands of S100A10. The manuscript could be more concise, in particular in the beginning of section 1, since S100A10 is not regulated by Ca2+. Introduction to EF-hand proteins and S100 could be more straightforward, since it had been reviewed many times as stated in abstract.

The main problem of the manuscript is the reference provided which are not always adequate or pertinent. The authors should carefully check all the cited reference and sometimes choose a more demonstrative citation. There seems to be completely mismatched in some sections. Here are few examples, but there are many more:

- line 170: ref 31 is not adequate for showing increased Ca2+ binding affinity

- line 213: ref 43 is not appropriate

- line 246: ref 60 is not appropriate

- line 309: ref 84-85 are not appropriate, especially 84

- line 346: ref 106 should be 108?

- line 351: ref 110 should be 111

Author Response

The manuscript by Okura et al reviews the functions and the binding partners of S100A10. The table 1 provides a comprehensive summary from the literature of all known ligands of S100A10. The manuscript could be more concise, in particular in the beginning of section 1, since S100A10 is not regulated by Ca2+. Introduction to EF-hand proteins and S100 could be more straightforward, since it had been reviewed many times as stated in abstract.

Authors comments–-Thank you for your comments and your careful review of the manuscript. We have gone through and reformatted all of the references so that the cited information is correct. 

The main problem of the manuscript is the reference provided which are not always adequate or pertinent. The authors should carefully check all the cited reference and sometimes choose a more demonstrative citation. There seems to be completely mismatched in some sections. Here are few examples, but there are many more:

All the references in the manuscript have been checked for accuracy. Unfortunately, in the initial submission, the reference numbers were mistakingly assigned to the wrong reference.

- line 170: ref 31 is not adequate for showing increased Ca2+ binding affinity

This sentence was changed to "The exception is S100A1, which upon covalent modification dramatically increases its Ca2+-binding affinity one-hundred-fold[38]." (Goch et al., 2005)

- line 213: ref 43 is not appropriate

ref 43 was changed to ref 50 (Zhu et al., 2021)

- line 246: ref 60 is not appropriate

ref 60 was changed to ref 67 (Surette et al., 2011)

- line 309: ref 84-85 are not appropriate, especially 84

ref 84-85 was changed to 92-93 (Tong et al., 2014;  Bagheri-Hosseinabadi et al., 2022)

- line 346: ref 106 should be 108?

ref 106 was changed to ref 111 (Swisher et al., 2010)

- line 351: ref 110 should be 111

ref 110 was changed to 115 (Sansico et al., 2021)

Round 2

Reviewer 2 Report

The authors have corrected the citation of refenreces in the text and have adressed all the remoarks.

Figures have been added to the manuscript. Unfortunately figure 1 and 3 are of lower quality than figure 4.

Author Response

Done.